# A Stochastic Model Based on Optimal Satellite Subset Selection Strategy for Smartphone Pseudorange Relative Positioning

**DOI:** 10.3390/s24082598

**Published:** 2024-04-18

**Authors:** Jian Deng, Huayin Wang, Shuen Wei, Aiguo Zhang

**Affiliations:** Department of Surveying and Remote Sensing Engineering, Xiamen University of Technology, Xiamen 361024, China; dengjian163@126.com (J.D.); huayinw@163.com (H.W.); kuuga592@163.com (S.W.)

**Keywords:** smartphone, stochastic models, carrier-to-noise ratio, pseudorange relative positioning

## Abstract

In order to overcome the limitations of traditional stochastic models for smartphones, we introduce a double-difference code pseudorange residual (DDPR)-dependent stochastic model based on an optimal satellite subset, with the goal of mitigating the constraints imposed by the quality of GNSS observations in smartphones on the accuracy and reliability of phone-based GNSS positioning. In our methodology, the satellite selection process involved considering the integrated carrier-to-noise density ratio (C/N0) index of both the reference station and the smartphone, enabling us to construct a satellite subset characterized by superior observation quality. Furthermore, by leveraging the optimal subset of satellites and incorporating the C/N0-dependent stochastic model, we could determine the approximate location of the terminal through pseudorange differential positioning. Subsequently, we estimated the DDPRs for all satellites and utilized these values as prior information to build a stochastic model of the observations. Our findings indicate that in occluded environments, the DDPR-dependent stochastic model significantly enhances positioning accuracy for both the Huawei Mate40 and P40 terminals compared to the C/N0-dependent model. Numerically, the improvements in the north (N), east (E), and up (U) directions were approximately 30%, 32%, and 34% for the Mate40, and 26%, 33%, and 24% for the P40 terminal. This suggests that the proposed DDPR-dependent stochastic model effectively identifies and mitigates large gross error signals caused by multipath and non-line-of-sight (NLOS) signals, thereby assigning lower weights to these problematic observations and ultimately enhancing positioning accuracy. Moreover, the weighting method involves minimal computations and is straightforward to implement, making it particularly suitable for GNSS positioning applications on smartphones in complex urban environments.

## 1. Introduction

Smartphones, as one of the most prevalent Global Navigation Satellite System (GNSS) positioning terminals in contemporary times, facilitate a myriad of navigation and positioning services integral to daily life. The advent of raw GNSS measurements from Android devices in recent years has paved the way for deriving more precise solutions through advanced positioning techniques. Pseudorange differential positioning, a kind of differential GNSS positioning technology, offers a time-efficient and highly effective alternative to carrier phase differential positioning. Particularly for GNSS smartphones, the absence of a need to resolve cycle slips and ambiguity, coupled with low computational power consumption and the simplicity of implementing a differential correction function, heralds a wide-ranging application prospect in the market. However, smartphones, unlike geodetic receivers, predominantly utilize linear polarization antennas and economical, low-power navigation chips. This makes their observations prone to noise and susceptible to multipath signals. Especially in urban environments characterized by dense buildings, the poor quality of observational data seriously affects the positioning accuracy and reliability of smartphones. In recent years, numerous studies have shown that smartphone positioning accuracy can achieve sub-meter levels with a single or dual-frequency PPP model [1,2,3]. However, the convergence time typically spans between 10 to 30 min, a duration primarily dependent on the surrounding environment [3,4,5]. In a real-time kinematics (RTK) mode, decimeter or even centimeter positioning accuracy can be realized in static open spaces under ideal conditions [6,7,8,9]. However, dynamic positioning in urban environments presents challenges due to observational noise and multipath influences, making ambiguity difficult to fix and reducing the average positioning accuracy to meter levels, thereby yielding suboptimal results [10,11,12]. Consequently, a discernible gap persists between smartphone-based location services and the high-precision positioning application demands of public life. The quality of observations, particularly pertaining to non-line-of-sight (NLOS) signals and multipath errors, remains a principal factor constraining the accuracy, timeliness, and reliability of smartphone GNSS positioning [3,6,13].

Ensuring the quality of GNSS observations in smartphones is pivotal for enhancing positioning performance. The stochastic model, which delineates the relative correlation and precision of GNSS observations, is crucial for subsequent optimal parameter estimation and quality control. Broadly, conventional observation stochastic models fall into two categories. The first encompasses classical prior weighting methods, where weights are determined based on empirical values or real-time observations. Examples include the elevation-dependent weighting model [14,15] and the carrier-to-noise density ratio (C/N0)-dependent weighting model [16,17]. The second category involves the posteriori variance estimation weighting model, which employs the correction number of the pre-adjustment to ascertain the weight. This includes methods like the Helmert variance component estimation and the minimum norm quadratic unbiased estimation methods [18,19]. Given these traditional models, many researchers have optimized them in tandem with practical applications, yielding notable results in GNSS positioning with geodetic receivers [20,21,22,23]. Nonetheless, smartphones exhibit distinct characteristics compared to geodetic receivers, with GNSS observations that are prone to noise and markedly affected by multipath and NLOS errors. In specific environments, like urban canyons, the low correlation between C/N0 and the corresponding satellite elevation means that the stochastic models for smartphones and professional geodetic receivers differ considerably [24,25].

Recent studies have indicated that the elevation-dependent weighting model might no longer be apt for smartphone GNSS observations [26,27,28]. The C/N0, which gauges signal reception quality, has been shown to be more optimized for smartphone positioning than the elevation-dependent model [8,24]. Recognizing the limitations of C/N0 in smartphone positioning, recent advancements include the elevation-C/N0 model, which merges both indicators (elevation and C/N0) with robust estimation principles [25], and the optimized C/N0-dependent stochastic model, which aptly describes smartphone GNSS observation quality [29]. These models have, to a degree, ameliorated smartphone positioning. However, the conventional C/N0-dependent stochastic model often fails to account for unmodeled errors like multipath and atmospheric delays, which are particularly significant for low-cost devices [30,31]. To achieve superior positioning performance with smartphones, there is a pressing need for more refined stochastic models. This study focuses on the construction of an optimal subset of satellites with integrated C/N0, which plays a crucial role in satellite observation accuracy. In obstructed environments, selecting satellites with superior observational quality becomes imperative. To address this, we propose synthesizing the C/N0 metric to identify a subset of satellites with higher-quality observations. Based on that, we introduce a novel approach leveraging the characteristics of the double-difference code pseudorange residual (DDPR). This approach entails the development of a DDPR-dependent stochastic model capable of detecting significant outliers attributed to multipath and NLOS effects. By identifying and mitigating these outliers, our model effectively reduces the corresponding observational weights, thereby enhancing positioning accuracy to a significant extent. Consequently, this methodological integration contributes to the overall robustness and reliability of satellite-based positioning systems in challenging environments.

The remainder of this paper proceeds as follows. In Section 2, we delve into the double-difference (DD) code pseudorange functional model. Drawing on the DDPR concept and utilizing experimental data, we examine the variation characteristics of a DDPR. Section 3 explores the correlation between the DDPR and the satellite-integrated C/N0 and elevation. We present a comparative analysis of methods for constructing the optimal satellite subset and detail the process of establishing a pseudorange relative positioning stochastic model for a smartphone-based optimal satellite subset. In Section 4, we contrast and analyze the positioning results of the optimized stochastic model and the C/N0-dependent model for Huawei P40 and Huawei Mate40 smartphones across diverse environments. Finally, Section 5 encapsulates the study’s findings and conclusions.

## 2. Positioning Model and Residual Analysis

### 2.1. Observation Model

Although many smartphones are currently capable of receiving dual-frequency or multi-frequency signals from the GNSS, when applied in real-world scenarios, limitations in the mobile terminal hardware or interferences from the surrounding environment can lead to situations where they are unable to continuously and completely receive all multi-frequency signals from the satellites. Consequently, this results in a smaller number of satellites with dual-frequency GNSS signals in each epoch, which is not conducive to positioning calculations. Furthermore, the stochastic model of observations examined in this study is applicable to both single-frequency and multi-frequency signals, with no fundamental distinction. Therefore, this paper takes single-frequency GNSS observational information as an example to study. In GNSS pseudorange differential relative positioning, DD observations are utilized to formulate an observation equation. Notably, the clock errors of the receiver and satellite in the observations are entirely negated through the inter-satellite and inter-station difference processes.

Given that a Global Positioning System (GPS) and Beidou Navigation Satellite System (BDS) are part of the code division multiple access (CDMA) signal system, each satellite within the system shares the same frequency, and the hardware delay of the receiver can be regarded akin to the receiver clock error, which is wholly eliminated during the inter-satellite difference process. Consequently, the fundamental observation equation model of GPS/BDS pseudorange relative positioning is expressed as
(1)∇∆P=∇∆ρ+∇∆I+∇∆T+∇∆M+∇∆O+∇∆ε
where ∇∆ denotes the DD operator, P is the pseudorange observation, ρ is the geometric distance between the satellite and the receiver, I and T are the ionospheric delay and tropospheric delay, respectively, M and O are the multipath and satellite orbit errors, respectively, and ε is the pseudorange observational noise. To facilitate equation solving, the approximate coordinates of the smartphone are assumed to be (x°, y°, z°), and the double-difference geometric distance ρ is linearized using Taylor expansion. The accuracy of the linearization process primarily depends on two factors: the error introduced by neglecting higher-order terms and the error due to the chosen approximation point deviating from the actual point. The former error is significantly smaller than the errors in pseudorange measurements caused by multipath effects and observation noise, and its impact on positioning results is minimal and can be neglected. To address the latter, the influence of this error is progressively reduced through multiple iterations during the computation, thereby enhancing the precision of linearization.

After linearization, the error equation can be expressed as
(2)V=BX−L
where X=[δx δy δz] represents the correction of the smartphone’s approximate coordinates, *V* denotes the correction of double-difference observations, and B is the corresponding coefficient matrix. L can be expressed as the following:(3)$$L=\nabla ∆P-\nabla ∆\rho^{0}-\nabla ∆I-\nabla ∆T-\nabla ∆M-\nabla ∆O-\nabla ∆\varepsilon$$
where $\rho^{0}$ is the geometric distance between the satellite and the receiver, calculated based on the smartphone’s approximate coordinates. In short-distance relative positioning, atmospheric and orbit errors can be neglected. If the weight matrix of the observations is W, representing the stochastic model of the observations, then according to the least squares principle, X can be calculated as
(4)$$X={\left(B^{T}WB\right)}^{-1}B^{T}WL$$
(5)$$W=Q^{-1}$$
(6)$$Q=\frac{1}{\sigma_{0}^{2}}\left[\begin{array}{cccc} a^{r}+a^{1} & a^{r} & \ldots & a^{r}\\ a^{r} & a^{r}+a^{2} & \ldots & a^{r}\\ \vdots & \vdots & \vdots & \vdots \\ a^{r} & a^{r} & \ldots & a^{r}+a^{n} \end{array}\right]$$
where $\sigma_{0}^{2}$ is the unit weight variance.$\;a^{r}={\sigma_{R}^{r}}^{2}+{\sigma_{S}^{r}}^{2}$, $a^{i}={\sigma_{R}^{i}}^{2}+{\sigma_{S}^{i}}^{2}\;(i=1,2,\ldots \ldots n,\;i\neq r)$, r, and i represent the reference satellite and non-reference satellite, respectively. ${\sigma_{R}^{i}}^{2}$ and ${\sigma_{S}^{i}}^{2}$ denote the variance in the observed values of each satellite for reference station R and smartphone S. Building upon the calculated X and incorporating the smartphone’s approximate coordinates (x°, y°, z°), one can ascertain the smartphone’s final coordinates, as shown in Equation (7).
(7)$$\left\{\begin{array}{c} x=x{}^{\circ}+\delta x\\ y=y{}^{\circ}+\delta y\\ z=z{}^{\circ}+\delta z \end{array}\right.$$

It can be seen that different variances in observations will result in different covariance matrices and different corresponding weights, and whether the weights are appropriate will directly affect the positioning performance. This paper aims to determine the weight of GNSS observations based on observational quality, minimizing the contribution of low-quality observations to enhance positioning accuracy.

### 2.2. Double-Difference Pseudorange Residual Analysis

The DD pseudorange is the basic observation of pseudorange relative positioning, and its quality directly affects the positioning. The DDPR is determined by subtracting the DD geometric distance between the satellite and the receiver from the DD pseudorange observation, and the equation is the following:(8)DDPR=∇ΔP−∇Δρ=∇ΔI+∇ΔT+∇ΔM+∇ΔO+∇Δε

It can be seen that the DDPR mainly includes the DD troposphere, ionosphere, satellite orbit, multipath error, and observational noise. This metric accurately reflects the quality of the observations when the terminal coordinates are precisely known. To enhance the quality of a reference station’s observations, a geodesic receiver can be used for relative positioning. This helps reduce both the multipath error and observational noise, thus improving the overall quality of observations by a reference station. Moreover, when the distance between the smartphone and the reference station is short, the influence of DD atmospheric and orbit errors can be disregarded. As a result, the DDPR primarily reflects the observational noise and multipath error of a smartphone. Figure 1 illustrates the DDPRs of each GPS/BDS satellite in a relatively open environment with an ultra-short baseline. Specifically, Figure 1a represents the Huawei P40 smartphone, while Figure 1b represents the geodesic receiver, and the two terminals were in the same location and correspond to the same reference station. From the figure, it is evident that the DDPRs exhibit significant differences between the smartphone and the geodesic receiver. The DDPRs of the geodesic receiver predominantly fall within the range of −2 m to 2 m, without any abnormally large values during testing. In contrast, the DDPRs of the smartphone show substantial variation among satellites and are primarily concentrated in the range of −20 m to 20 m. Some epochs even display intermittent jumps, reaching nearly 100 m. Preliminary analysis suggested that smartphones use linear polarized antennas and low-cost, low-power navigation chips, their observation noise is high, and compared with geodesic receivers, they are more susceptible to errors such as multipath effects or NLOSs, resulting in large DDPRs. Consequently, effective detection and elimination of signals with gross errors or reduction in the observational weight of poor-quality observations can potentially enhance the positioning accuracy.

However, in practical applications, precise terminal coordinates are often unknown, making it challenging to obtain an accurate DDPR to assess observational quality. To overcome this limitation, this study proposes selecting a subset of satellites with the most reliable observational quality for positioning, thus obtaining an initial terminal position. These chosen satellites are less susceptible to multipath, NLOS, and other errors, yielding relatively dependable positioning results. Based on this approach, the DDPRs of all visible satellites can be estimated. Although the residuals are not accurate enough due to the approximate position of the terminal, they can still reflect large gross errors such as multipath ones and NLOSs in the observations to a certain extent. Finally, the stochastic model of observations was constructed by using DDPRs to further improve positioning accuracy.

## 3. DDPR-Dependent Model Based on Optimal Satellite Subset

The key purpose of this method is to select the satellite with better observation quality from all visible satellites. Two main criteria are generally used to evaluate the quality of observations: the satellite elevation and the C/N0 value. Previous research has demonstrated that single-differenced pseudorange residuals have a stronger correlation with C/N0 values than satellite elevations, particularly for smart devices [4]. When it comes to relative positioning, the basic observation is typically in the form of DD observation. The reference satellite observations included in each DD observation are the same; hence, the quality difference primarily depends on the non-reference satellite observations of the reference station and the smartphone. Given that the C/N0 value can more accurately reflect the quality of satellite observations, an integrated C/N0—which is defined as the sum of the C/N0 of the non-reference satellite corresponding to the reference station and the smartphone—is used as the quality evaluation index of the DD code pseudorange observations.

The data of the Huawei P40 smartphone in the ultra-short baseline described in the above section were further analyzed. Two satellites, G07 and C09 in the GPS and BDS systems, respectively, which have a long viewing time and a large range of elevation variation, were selected. Figure 2 illustrates the correlation between the DDPR and the satellite-integrated C/N0 and elevation.

A larger DDPR indicates a poorer quality of double-difference observations. From the figure, it is evident that when the DDPR fluctuates within the range of −10 m to 10 m, the corresponding integrated C/N0 is primarily concentrated within the range of 70 to 90 dB-Hz. Particularly, when the DDPR becomes abnormally large, for example, around the 2000th and 8000th epochs of satellite G07 and around the 3000th and 6000th epochs of satellite C09, the corresponding integrated C/N0 also suddenly changes and is far less than the normal value. Conversely, when comparing the satellite elevation with the DDPR, Figure 2 reveals that the correlation between the DDPR and satellite elevation is not obvious on the whole. Furthermore, low-elevation satellites, compared to high-elevation satellites, have a higher probability of DDPR outliers.

The correlation between DDPR and integrated C/N0, as well as the elevation of all satellites, was further analyzed. The DDPRs were sorted in ascending order of the satellite elevation angle and integrated C/N0, respectively, and the mean of absolute values of each interval of 1 degree and 1 dB-Hz were computed. Figure 3 shows the mean values of absolute DDPRs at different elevation angles and integrated C/N0 values. As can be seen from Figure 3a, the mean value of the DDPR fluctuates greatly in the sequence of satellite elevation changes. In general, the DDPR tends to decrease with the increase in satellite elevation, but the whole trend line is not smooth enough and the correlation is not obvious. In Figure 3b, the integrated C/N0 values above 80 dB-Hz correspond to mean DDPR values within 3 m, indicating better observational quality. Overall, there is a consistent decrease in the DDPR with increasing integrated C/N0, demonstrating a smooth trend line and strong correlation between the two variables.

As analyzed in the previous section, the integrated C/N0 and satellite elevation can to some extent indicate the quality of satellite observations, and the satellite selection according to different quality evaluation indicators will lead to different positioning effects. This study attempts to develop three satellite selection schemes based on quality assessment indicators. Scheme 1 is developed using the integrated C/N0. Scheme 2 utilizes the satellite elevation as its basis. Scheme 3 combines both the satellite C/N0 and satellite elevation for its development. In Scheme 3, the influences of both the integrated C/N0 and satellite elevation were taken into account. To scale both datasets to the same range and remove the impact of dimensions, the corresponding integrated C/N0 and satellite elevation were initially normalized. Subsequently, a comprehensive quality assessment indicator CQ, which combines the C/N0 and satellite elevation, was formulated, with its expression being
(9)$$CQ=a\times \hat{c/N0}+b\times \hat{ele}$$
where $\hat{c/N0}$ and $\hat{ele}$ represent the normalized integrated C/N0 and satellite elevation, respectively, while a and b are the corresponding weight coefficients. Given that the integrated C/N0 is more reflective of the quality of satellite observations than the satellite elevation, this paper set the values of a and b to 0.6 and 0.4, respectively. In each satellite selection scheme, we will select the N satellites with the optimal quality evaluation indicators to participate in the positioning calculation. The value of N can be determined based on the surrounding environment of the smartphone and the total number of satellites that can be observed. For example, in an open environment, the value of N can usually be set between 12 and 15, while in an obstructed environment, it can be set between 8 and 10.

For comparing the performance of the three mentioned schemes, observational data were statically collected for nearly 3 h using Huawei Mate40 and Huawei P40 smartphones in a relatively open environment, using a geodetic receiver as the reference station. Utilizing the gathered experimental data, single-frequency pseudorange relative positioning was conducted using the three aforementioned schemes, and the positioning results for each epoch were compared with the reference values. Figure 4 illustrates the distributions of positioning errors on the horizontal plane for each scheme. As can be seen, for the Huawei Mate40 smartphone, when using Scheme 1 for positioning, the distribution of error points is more concentrated in the east (E) direction than in the north (N) direction. Overall, the horizontal positioning errors are relatively centralized, predominantly falling within the range of [−5 m, 5 m]. In comparison to Scheme 1, Scheme 2 exhibits a noticeably larger range of positioning errors in both the E and N directions, with the distribution primarily concentrated within the [−10 m, 10 m] range. The interval of concentrated point error distribution in Scheme 3 falls between that of Schemes 1 and 2.

Table 1 presents the root mean square (RMS) of the positioning results for each scheme. The RMS is mainly used to measure the deviation between a set of observed values and the true values, and it can well reflect the overall accuracy of measurements. From Table 1, it is evident that Scheme 1 exhibits the smallest RMS values in both the N and E directions across the three schemes, with Scheme 3 coming next. On the whole, in the horizontal direction, the positioning RMS values for the three schemes are 3.55 m, 6.11 m, and 4.96 m, respectively. This demonstrates that the satellite selection scheme based on the integrated C/N0 is capable of selecting a relatively optimal subset of satellites, resulting in better positioning outcomes. Conversely, the satellite selection scheme based on elevation yields the lowest positioning accuracy. Satellite selection Scheme 3 based on the C/N0 and satellite elevation, which takes into account both factors, has a positioning effect that lies between the two.

For the Huawei P40 smartphone, Scheme 1 also demonstrates a better concentration of position errors for each epoch compared to the other two schemes. The RMS values of the three schemes in planar positioning are 4.19 m, 5.70 m, and 5.41 m, respectively, showing a consistent pattern with the Huawei Mate40 smartphone. Therefore, compared to satellite elevation, the integrated C/N0 is more indicative of the quality of GNSS observational values for a smartphone, and a satellite selection scheme based on integrated C/N0 can select satellites of higher quality to ensure better positioning accuracy.

According to the above analysis, this paper proposes a DDPR-dependent stochastic model based on the optimal satellite subset with an integrated C/N0. For a visual representation, the algorithm flowchart is provided in Figure 5. The steps to establish the stochastic model are outlined as follows.

First, combined with the C/N0 of both the reference station and the smartphone, the observational quality evaluation index, namely the integrated C/N0, was calculated. Based on this, the optimal satellite subset for the pseudorange differential positioning solution was determined.

Secondly, utilizing the optimal subset of satellites, coupled with Equation (10) defining C/N0 variance, the C/N0-dependent stochastic model was established, and the approximate terminal position can be derived by pseudorange differential positioning.
(10)$${\sigma_{R/S}^{i}}^{2}=\hat{a}+\hat{b}\times 10^{-\frac{{\left(C/N_{0}\right)}_{i}}{10}}$$
where σ represents the pseudorange noise. C/N0 represents the carrier-to-noise density ratio for each satellite, which can be directly obtained from the smartphone’s observation values. Parameters $\hat{a}$ and $\hat{b}$ needed to be determined for each piece of equipment at different frequencies and constellations. For further detail on the calibration of parameters $\hat{a}$ and $\hat{b}$, refer to the work of Liu et al. [32,33].

Finally, the DDPRs for each satellite were estimated. While the residual is influenced by the terminal’s approximate position, it can still signify the quality of the DD observations, particularly gross errors like multipath errors and NLOSs. By taking the reciprocal of its absolute value as the weight factor, an observational stochastic model was formulated, expressed as the following:(11)$$W=diag(w_{1\;}{\;w}_{2\;}\cdots \;w_{m\;})$$
(12)$$w_{i\;}=\frac{1}{\left|{DDPR}_{i}\right|}$$

In essence, this was achieved by diminishing the weight of observations with large pseudorange residuals, thereby mitigating their impact on the positioning results.

## 4. Results and Discussion

The relative positioning experiments for smartphones were conducted in both open and occluded environments. The experimental setup is depicted in Figure 6. The reference station, equipped with a geodetic receiver, was situated in an open environment on a rooftop, as illustrated in Figure 6a. The smartphones used for this experiment were the Huawei Mate40 and Huawei P40. In the open environment, both smartphones were placed on the roof of a high-rise building, ensuring open and unobstructed surroundings, as shown in Figure 6b. They formed an ultra-short baseline of nearly 5 m with the reference station. Conversely, in the occluded environment, the smartphones were positioned adjacent to a high-rise building. This setup resulted in a short baseline of over ten meters with the reference station. In this setting, the satellite signals for the smartphones were prone to blockages and interference, leading to multipath errors and NLOSs, as depicted in Figure 6c. Static GNSS observation data were gathered for approximately 2 h in each environment, with a sampling interval set at 1 s. Subsequently, the baseline reference values were derived using high-precision static data processing software.

### 4.1. Open Environment

In the experiment, GPS and BDS satellites were chosen. Figure 7 displays the distributions and counts of the observed satellites in the open environment. It is evident that throughout the experiment, the satellite count in the northern direction was relatively sparse. The combined count of the BDS/GPS satellites per epoch never dropped below 16, and consistently remained between 24 and 27 for the majority of the time.

The relative positioning performance for Mate40 and Huawei P40 smartphones were derived using code pseudorange observations for each epoch. These were then juxtaposed against known benchmark coordinates. Figure 8 delineates the positioning errors in the N, E, and up (U) directions using both the C/N0-dependent stochastic mode and the DDPR-dependent mode. The positioning errors in the N and E directions for both modes predominantly fall within the [−5 m, +5 m] range, with marginal differences between them. However, the U direction errors are more pronounced, mainly spanning the [−10 m, +10 m] range. Observational errors, such as multipath errors, caused anomalies with substantial errors in certain epochs across all three directions. Broadly speaking, the positioning precision with both stochastic modes is comparable. Nevertheless, during specific epochs, like the 3000 to 4000 epoch span for the Huawei Mate40, the presence of satellites with inferior observational quality leads to a noticeable surge in positioning errors. During these instances, the DDPR-dependent stochastic model, which reduced the weight of satellite observations with significant pseudorange residuals, showcased superior positioning accuracy compared to the C/N0-dependent stochastic mode.

Table 2 presents the RMS values of the positioning results derived from the various stochastic models. The data clearly show that, only for the Huawei P40 in the N direction, the RMS value based on the DDPR-dependent stochastic model is about 0.3 m higher than that of the C/N0-dependent stochastic mode, while in other directions, the former is smaller than that of the latter. In general, there is little difference in the RMS values obtained by the two stochastic models, but the overall positioning performance based on the DDPR-dependent model is slightly better than that of the C/N0-dependent model.

The final column in Table 1 also enumerates the positioning performance when utilizing the optimal satellite subset with the C/N0-dependent stochastic mode. It is obvious that its positioning accuracy is slightly inferior to that of the full satellite positioning (as indicated in the third column of Table 2). This discrepancy primarily stems from the high-quality signals received from all satellites in an open environment. When juxtaposed against full constellation positioning, the optimal subset satellite positioning employs fewer satellites, resulting in a larger DOP value. Consequently, the corresponding RMS is also relatively higher. In essence, the optimal subset satellite does not offer a pronounced advantage in open environments.

### 4.2. Occluded Environment

The distributions and counts of observed satellites in the occluded environment are depicted in Figure 9. It is evident that satellites predominantly cluster in the southwest direction. The combined count of GPS/BDS satellites observed in each epoch is never less than 14, with the count consistently hovering between 21 and 24 for the majority of the time. The surrounding buildings’ occlusion resulted in a reduction of 4–6 satellites on average for the smartphone, compared to the open environment, leading to significant fluctuations in satellite counts across epochs.

Single-frequency code pseudorange observations were employed to calculate the relative positioning of the Huawei Mate40 and Huawei P40 smartphones per epoch, which were then juxtaposed against known benchmark coordinates. Figure 10 illustrates the time series of positioning errors using different stochastic models. In the occluded environment, smartphones were susceptible to errors like multipath ones and NLOSs, degrading the quality of observations. The relative positioning accuracy per epoch is diminished, with errors in the N and E directions primarily spanning the [−10 m, +10 m] range. However, errors in the U direction are more pronounced, clustering within the [−20 m, +20 m] range. A comparison of the two stochastic models, as visualized in Figure 10, revealed that the error fluctuation range in the N, E, and U directions for the DDPR-dependent model (based on the optimal subset) was markedly reduced compared to the C/N0-dependent stochastic mode, especially during specific periods. For instance, during the initial 2000 epochs for the Huawei P40 smartphone in the N direction, the C/N0-dependent mode’s positioning accuracy frequently exceeded 10 m. In contrast, the DDPR-dependent model proposed in this study managed to maintain accuracy within 5 m.

In Figure 11, the cumulative distribution function (CDF) for positioning error indicated that for the Huawei Mate40 and Huawei P40 smartphones, the proportion of horizontal positioning accuracy within 5 m using the DDPR-dependent model stood at 57% and 60%, respectively. This is a significant improvement over the C/N0-dependent mode’s respective 23% and 42%. Similarly, in the vertical direction, the DDPR-dependent model achieved 41% and 40% accuracy within 5 m, respectively, outperforming the C/N0-dependent mode’s respective 27% and 30%. It is evident that the positioning accuracy comparisons between the two smartphones using the two stochastic models are consistent. In the occlusion environment, the C/N0 of certain satellites failed to accurately represent the impact of multipath errors and NLOSs in observations, revealing the C/N0-dependent model’s inherent limitations. The proposed DDPR-dependent model, based on the optimal satellite subset’s terminal approximate position, can identify significant gross errors caused by multipath and NLOS issues. By reducing the weight of corresponding observations, positioning accuracy is enhanced to a degree.

Table 3 consolidates the positioning result statistics for both stochastic models. For the Huawei Mate40 smartphone, the RMS of the DDPR-dependent model decreased by approximately 2 m in the N and E directions, and notably, by nearly 6 m in the U direction, when compared to the C/N0-dependent model. The positioning accuracy across these directions improved by 30%, 32%, and 34%, respectively. Similarly, for the Huawei P40 smartphone, the positioning accuracy in all three directions also saw an enhancement of 26%, 33%, and 24%, respectively. Moreover, a comparison between the whole constellation-based positioning (column 3) and the optimal satellite subset-based positioning (last column) revealed that even when the same C/N0-dependent stochastic model was utilized, the RMS values derived from the optimal satellite subset outshined those from the whole constellation in the N, E, and U directions for both smartphones. The comparison results differ from those in the above open environment, mainly due to the influence of significant errors such as multipath ones and NLOSs affecting some satellite signals in the harsh occluded environment. By selecting the satellite with the highest integrated C/N0 and a lower likelihood of significant errors, the impact of poor-quality observations on positioning performance is minimized to some extent. Simultaneously, improved positioning outcomes aid in constructing more accurate DDPRs, facilitating gross error detection and weight determination and further enhancing positioning efficacy.

## 5. Conclusions

A stochastic model plays a crucial role in determining the accuracy of a positioning model solution. However, the traditional C/N0-dependent stochastic model often fails to account for the impacts of unmodeled errors such as multipath and atmospheric delays. Leveraging the concept of a DDPR and experimental data, this paper delved into the correlation between DDPR, integrated C/N0, and satellite elevation. Subsequently, we introduced a DDPR-dependent stochastic model based on the optimal satellite subset. Real-world datasets collected from a Huawei P40 and a Huawei Mate40 were utilized to validate the proposed method. The conclusions of this study are the following:The DDPR effectively reflects the quality of observations. While the correlation between the DDPR and satellite elevation is not significant, it exhibits a strong correlation with the integrated C/N0.In occluded environments, a satellite subset boasting superior observational quality can be selected using the integrated C/N0. The positioning accuracy, when relying on this optimal satellite subset, markedly surpasses that derived from the entire constellation.In open settings, the merits of a DDPR-dependent stochastic model in pseudorange relative positioning are not markedly superior to the C/N0-dependent stochastic model. The experimental results from both smartphones indicate comparable positioning accuracy between the two models. However, in an occluded setting, the former model proficiently diminishes the weight of inferior-quality observations. This leads to enhanced positioning accuracy for both smartphones. Specifically, the Huawei Mate40 exhibited improvements of 30%, 32%, and 34% in the N, E, and U directions, respectively. The Huawei P40 also showcased enhancements of 26%, 33%, and 24% in the same respective directions.

A DDPR, when based on an optimal satellite subset, can somewhat capture the effects of significant gross errors, such as multipath ones and NLOSs, in its observations. The resultant stochastic model can mirror the impact of unmodeled errors. This model, with its ease of implementation and low computational demands, shows promise for enhancing GNSS positioning in smartphone navigating in complex urban environments. However, the selection of the optimal satellite subset remains paramount in model construction. Future endeavors will holistically weigh factors such as satellite elevation, C/N0, and satellite spatial distribution to refine satellite selection strategy.

## Figures and Tables

**Figure 1 sensors-24-02598-f001:**
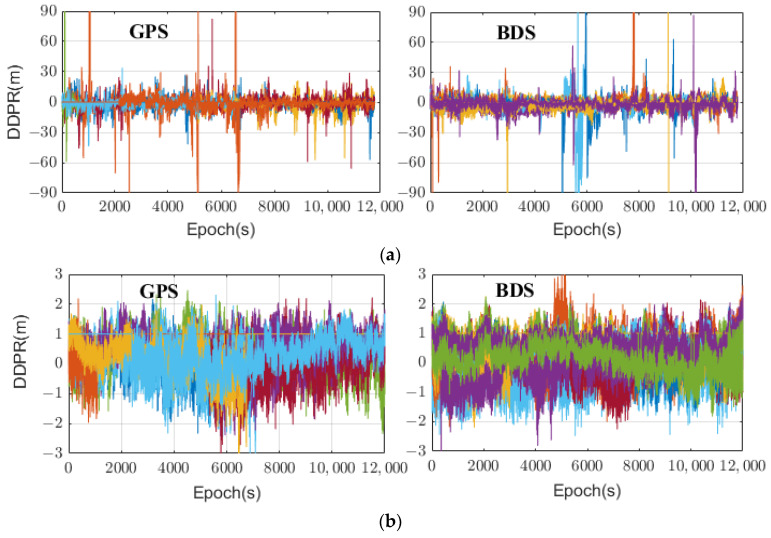
DDPRs of different terminals in a relatively open environment. (**a**) Huawei P40 smartphone. (**b**) Geodesic receiver.

**Figure 2 sensors-24-02598-f002:**
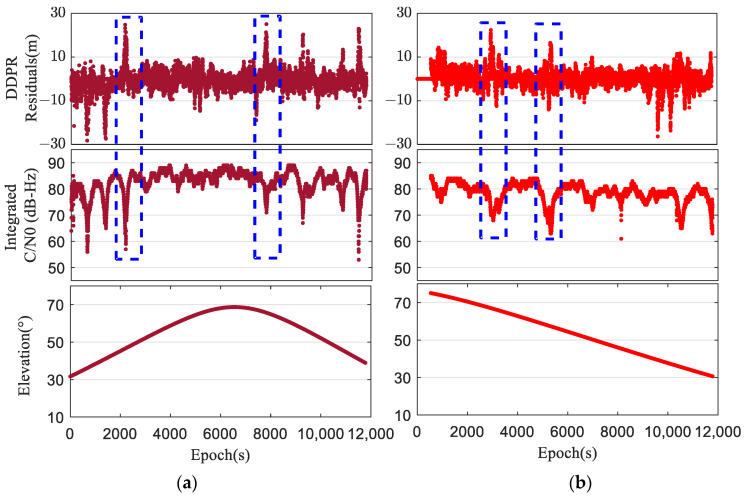
DDPRs against the satellite C/N0 and elevation. (**a**) Satellite G07. (**b**) Satellite C09.

**Figure 3 sensors-24-02598-f003:**
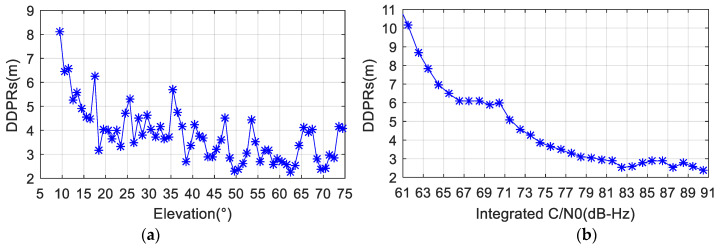
Mean values of absolute DDPRs changing with (**a**) elevation angles and (**b**) the integrated C/N0 for a smartphone.

**Figure 4 sensors-24-02598-f004:**
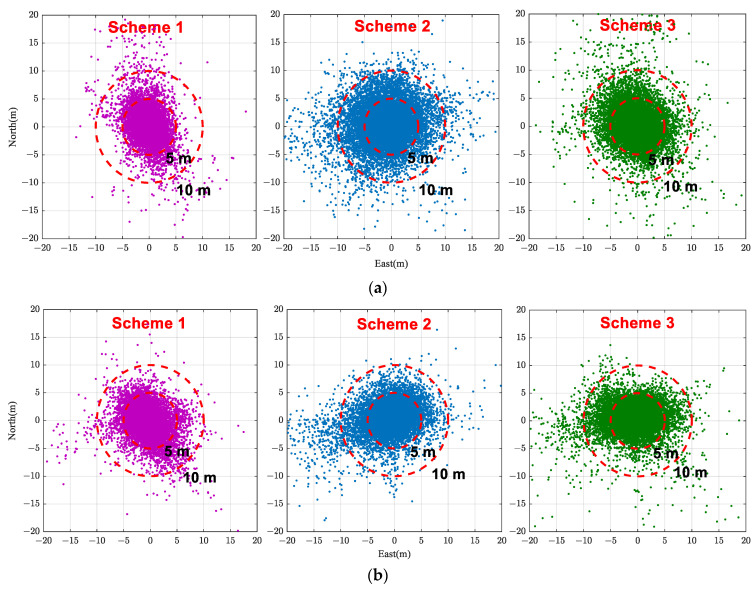
Distributions of positioning errors on the horizontal plane for each satellite selection scheme. (**a**) Huawei Mate 40 smartphone. (**b**) Huawei P40 smartphone.

**Figure 5 sensors-24-02598-f005:**
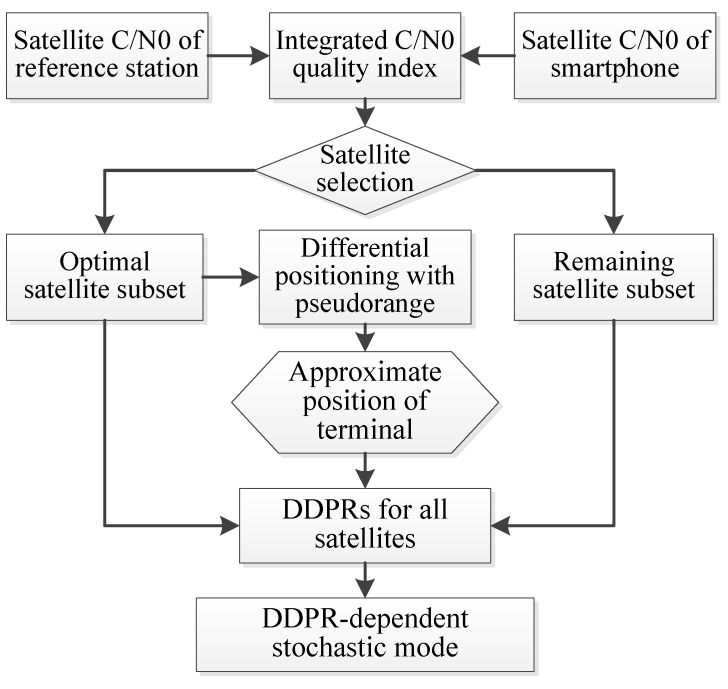
Flowchart of DDPR-dependent model based on optimal satellite subset.

**Figure 6 sensors-24-02598-f006:**
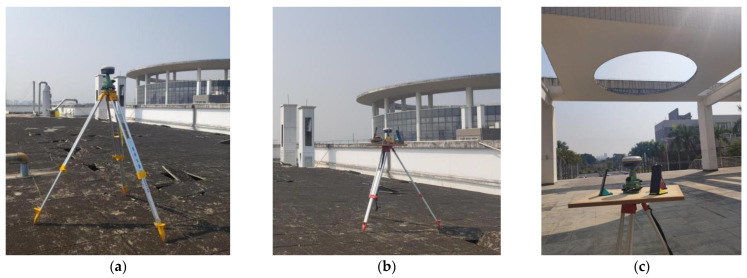
Experimental scenarios. (**a**) Reference station. (**b**) Smartphones in an open environment. (**c**) Smartphones in an occluded environment.

**Figure 7 sensors-24-02598-f007:**
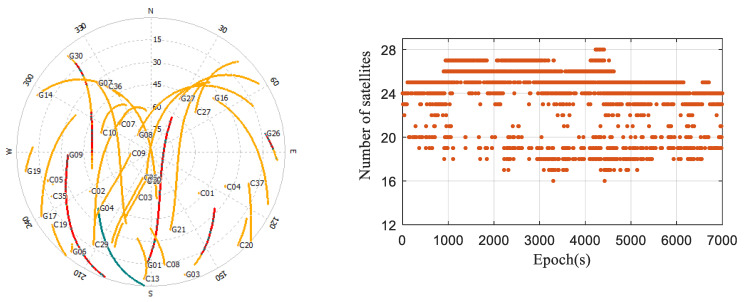
Distributions and counts of the observed satellites in the open environment.

**Figure 8 sensors-24-02598-f008:**
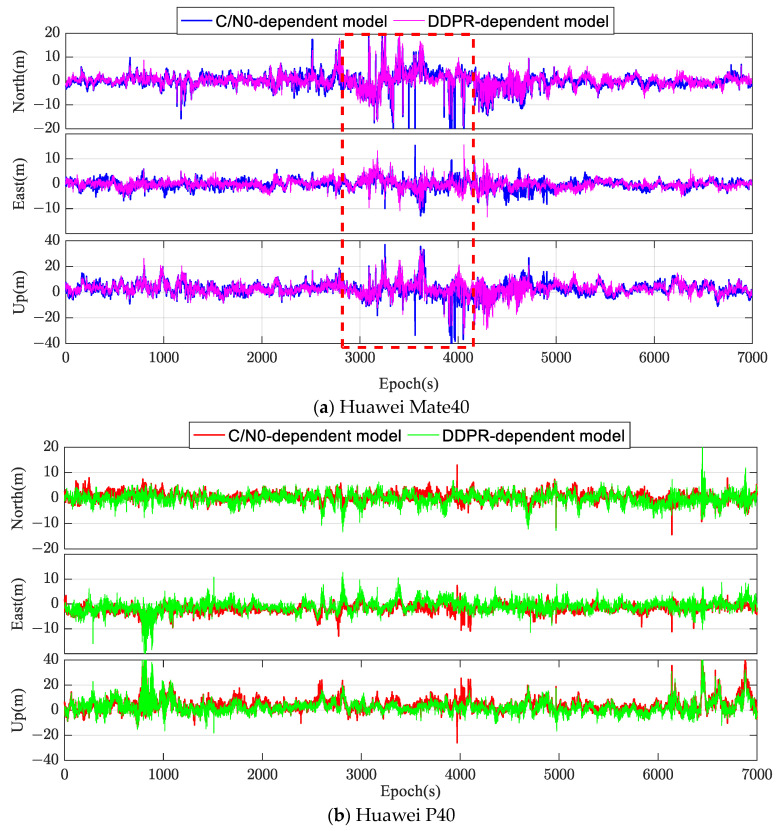
Time series of the positioning errors with different stochastic models in the open environment.

**Figure 9 sensors-24-02598-f009:**
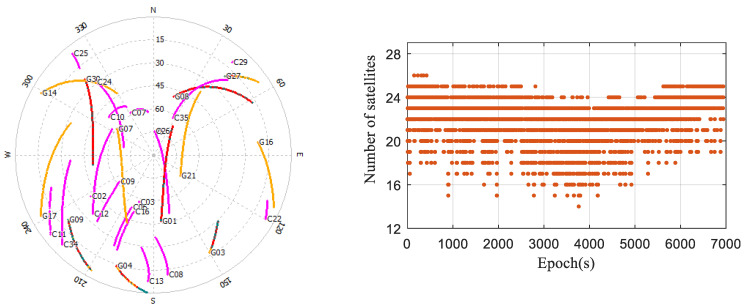
Distributions and counts of the observed satellites in the occluded environment.

**Figure 10 sensors-24-02598-f010:**
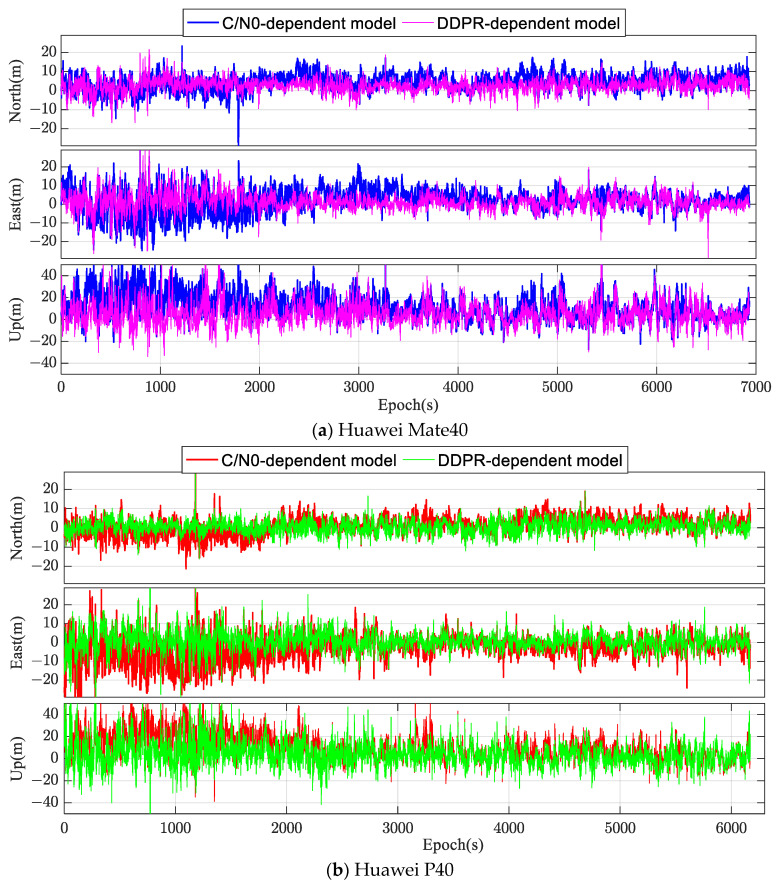
Time series of the positioning errors with different stochastic models in the occluded environment.

**Figure 11 sensors-24-02598-f011:**
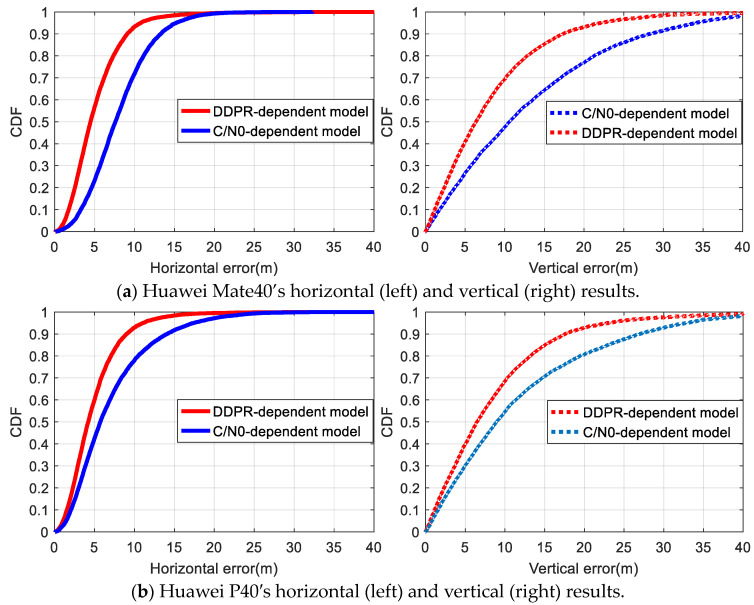
CDFs of positioning error based on the two models in the occluded environment.

**Table 1 sensors-24-02598-t001:** RMS values of positioning results with different satellite selection schemes (unit: meter).

Selection Scheme	Huawei Mate40			Huawei P40		
	N	E	Horizontal	N	E	Horizontal
Integrated C/N0	2.90	2.05	3.55	2.87	3.06	4.19
Satellite elevation	3.89	4.71	6.11	2.88	4.92	5.70
Elevation-C/N0	3.71	3.29	4.96	2.88	4.58	5.41

**Table 2 sensors-24-02598-t002:** RMS values of positioning results with different stochastic models in an open environment.

Device	Direction	C/N0-Dependent (m)	DDPR-Dependent (m)	Accuracy Improvement	C/N0-Dependent (Optimal Subset) (m)
Mate40	N	3.36	3.14	6%	3.17
	E	1.93	1.80	7%	2.19
	U	5.85	5.72	2%	6.28
P40	N	1.95	2.24	−15%	2.91
	E	2.58	2.56	1%	3.06
	U	6.61	6.12	7%	6.42

**Table 3 sensors-24-02598-t003:** RMS values of positioning results with different stochastic models in the occluded environment.

Device	Direction	C/N0-Dependent (m)	DDPR-Dependent (m)	Accuracy Improvement	C/N0-Dependent (Optimal Subset) (m)
Mate40	N	6.01	4.19	30%	4.25
	E	6.67	4.50	32%	4.72
	U	16.75	11.01	34%	12.34
P40	N	4.54	3.36	26%	3.68
	E	7.45	4.98	33%	5.53
	U	15.66	11.90	24%	14.19

## Data Availability

Due to restrictions, data are only available upon request.

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
