# Peer review of "A Stochastic Model Based on Optimal Satellite Subset Selection Strategy for Smartphone Pseudorange Relative Positioning"

_sensors, 2024, doi:10.3390/s24082598_

Round 1
Reviewer 1 Report
Comments and Suggestions for Authors
The paper is well written and backed up by several experimental and theoretical results and studies. It would be better if the authors can use more texts to describe how the receiver's coordinates is calculated from Android reading in detail.
Reviewer 2 Report
Comments and Suggestions for Authors
The paper introduces a novel approach to address limitations in traditional stochastic models for smartphone GNSS positioning. The proposed method, termed Double-Difference Code Pseudo-Range Residual (DDPR) dependent stochastic model, aims to enhance accuracy and reliability despite constraints imposed by GNSS observation quality on smartphones. This is achieved by optimizing satellite selection based on carrier-to-noise density ratio (C/N0) indices, constructing a superior satellite subset. By incorporating this subset and a C/N0-dependent stochastic model, the methodology estimates terminal locations via pseudo-range differential positioning. Notably, DDPR-dependent modeling significantly improves accuracy in occluded environments for Huawei Mate and P terminals compared to C/N0-dependent models. The model effectively identifies and mitigates errors caused by multipath and NLOS, enhancing accuracy by assigning lower weights to problematic observations. Additionally, the proposed weighting method is computationally efficient and easily implementable, making it suitable for smartphone GNSS positioning in complex urban settings.
The introduction does not highlight the details about the novelty of the presented methodology. The authors should underline which are the pros and cons of the presented work, with a proper analysis and a clear review of the novelties and scientific relevance compared to the state of the art.
Throughout the paper, all the acronyms have to be explained (e.g. DD, CDMA, GPS, etc.). Furthermore, the punctuation must be double-checked all along the text.
All the formulas along the text must be verified, and all the variable in each equation should be cited properly in the text (e.g. in equation (1), the pseudo-range observation noise is 𝜀𝑝 while in the text it is cited as 𝜀).
In the text, the authors are referring to the "rover's approximate coordinates". To which rover are the authors referring? The paper is related to a positioning system for a smartphone, and no rover is mentioned before.
Equation (2): variable V is not mentioned.
Equation (9): please do not use parameters that have been previously defined in the paper with other meaning.
Comments on the Quality of English LanguageThe english language could be enhanced. Specifically, the authors should take care of punctuation and may reformulate some sentences in order to make the text more clear.
Reviewer 3 Report
Comments and Suggestions for Authors
I liked the very detailed analysis of the work on the use of modern means to improve the performance of smartphones, especially in an urban environment. The authors have developed an optimized stochastic model for evaluating the quality of observations. In this study, an attempt was made to compare three satellite selection schemes based on quality assessment indicators: one of them is based on integrated C/N0 (scheme 1), another is based on satellite altitude (scheme 2) and the third is based on a combination of C/N0 satellite altitude and satelite altitude (Scheme 3) (Fig. 4). Moreover, the second scheme turned out to be the least accurate.
These schemes are compared in open and closed environments. It seems to me that the advantage of the work is the separation of open and closed environments. It should be added that the authors separately consider horizontal and vertical errors (Fig. 11). The authors paid special attention to smartphones in a closed urban environment, the quality of this model is confirmed by the fact that the improvements amount to about 30 percent. Therefore, this article can already be recommended for publication in the journal cencors. From a purely methodological point of view, all three of these methods of separate research in a rather complex practical task are productive.
However, there are the following comments on the somewhat confusing presentation of the material.
1) I will start with linearization in formulas (2), (3), for which it is desirable to specify the accuracy of the linear approximation.
2) On the one hand, the authors say that with a small distance between the smartphone and the reference station, atmospheric and orbit errors can be neglected. On the other hand, it is argued that low-lying satellites have a higher probability of emissions compared to high-lying satellites. It seems to me that this statement should be clarified taking into account the various special cases considered in the work.
3) I would also like to clarify the conditions under which formula (9) containing the exponent can be used. Unfortunately, it was not possible to find a clear definition of the parameter N_0.
